# Predictors of Breakthrough SARS-CoV-2 Infection after Vaccination

**DOI:** 10.3390/vaccines12010036

**Published:** 2023-12-28

**Authors:** Sharon Walmsley, Majid Nabipoor, Leif Erik Lovblom, Rizani Ravindran, Karen Colwill, Alison McGeer, Roya Monica Dayam, Dorin Manase, Anne-Claude Gingras

**Affiliations:** 1Division of Infectious Diseases, Department of Medicine, University Health Network, Toronto, ON M5G1L7, Canada; rizani.ravindran@uhn.ca; 2Department of Medicine, University of Toronto, Toronto, ON M5S1A1, Canada; 3Biostatistics Department, University Health Network, Toronto, ON M5G1L7, Canada; nabipoor@thebru.ca (M.N.); erik.lovblom@thebru.ca (L.E.L.); 4Lunenfeld-Tanenbaum Research Institute, Mount Sinai Hospital, Sinai Health, Toronto, ON M5G1X5, Canada; colwill@lunenfeld.ca (K.C.); rdayam@lunenfeld.ca (R.M.D.); gingras@lunenfeld.ca (A.-C.G.); 5Mount Sinai Hospital, Sinai Health, Toronto, ON M5G1X5, Canada; amcgeer@mtsinai.on.ca; 6DATA Team, University Health Network, Toronto, ON M5G1L7, Canada; dorin.manase@uhn.ca; 7Department of Molecular Genetics, University of Toronto, Toronto, ON M5S1A1, Canada

**Keywords:** COVID, SARS-CoV-2, vaccine, breakthrough, serology, booster, antibody

## Abstract

The initial two-dose vaccine series and subsequent booster vaccine doses have been effective in modulating SARS-CoV-2 disease severity and death but do not completely prevent infection. The correlates of infection despite vaccination continue to be under investigation. In this prospective decentralized study (n = 1286) comparing antibody responses in an older- (≥70 years) to a younger-aged cohort (aged 30–50 years), we explored the correlates of breakthrough infection in 983 eligible subjects. Participants self-reported data on initial vaccine series, subsequent booster doses and COVID-19 infections in an online portal and provided self-collected dried blood spots for antibody testing by ELISA. Multivariable survival analysis explored the correlates of breakthrough infection. An association between higher antibody levels and protection from breakthrough infection observed during the Delta and Omicron BA.1/2 waves of infection no longer existed during the Omicron BA.4/5 wave. The older-aged cohort was less likely to have a breakthrough infection at all time-points. Receipt of an original/Omicron vaccine and the presence of hybrid immunity were associated with protection of infection during the later Omicron BA.4/5 and XBB waves. We were unable to determine a threshold antibody to define protection from infection or to guide vaccine booster schedules.

## 1. Introduction

The mRNA COVID vaccines have demonstrated excellent efficacies against severe infection and death from the SARS-CoV-2 ancestral (Wuhan) strain [1,2,3,4,5]. Natural infection is partly protective against re-infection [6,7,8]. To date, the degree of protection that antibodies conferred by COVID-19 vaccines and/or natural infection against re-infection or transmission, especially with emerging variants [9,10], cannot be quantified. Investigations consider vaccine-induced antibody titers as a marker of immune protection from infection or severe COVID-19 illness [11,12,13] but no antibody threshold at which to recommend booster dosing after the primary vaccine series has been determined [14]. Some studies report threshold antibody levels required for 50 or 70% protection against the ancestral and early SARS CoV-2 variant strains but note that the risk of infection changes gradually with the antibody titers [15,16,17,18]. The World Health Organization (WHO) has established standardized units for antibodies against SARS-CoV-2 (BAU/mL; binding assay); however, reported titers may be affected by the assays used, the vaccinated populations studied and the prevalent viral variant strains [16], making comparisons between studies difficult.

Population-based studies show waning vaccine effectiveness with time and lower activity against the emergent SARS-CoV-2 variant strains [6,11,19,20,21,22]. Breakthrough infection rates are rising. The proportion of the population infected with COVID-19 has increased and, in Canada, is approaching 80% [23]. However, the vaccines have modulated disease such that the rates of hospitalization and death have decreased [20,21,24].

In response, bivalent vaccines were developed and recommended for subsequent booster doses [25,26,27]. Cohort studies have shown increased neutralizing antibody responses to bivalent vaccines relative to boosters with the original vaccine brands against both the ancestral strain and SARS-CoV-2 variants [28,29,30]. Population-based studies using test-negative case–control designs have shown increased vaccine effectiveness in preventing hospitalization and symptomatic infection due to SARS-CoV-2 variants with bivalent booster dosing in multiple geographic settings [25,31,32,33,34,35,36].

The STOPCoV study is an ongoing decentralized longitudinal study aimed to compare COVID-19 antibody levels in those aged ≥ 70 years relative to a cohort aged 30–50 years [37]. In this analysis, we aimed to assess the predictors of breakthrough SARS-CoV-2 infection in this well-characterized and highly vaccinated population.

## 2. Methods and Materials

The main STOPCoV study with complete methods has been published [37], with the full protocol available on the study website www.stopcov.ca, accessed on 20 December 2023. The trial is registered on Clinicaltrials.gov; identifier NCT05208983. In brief, STOPCoV is a decentralized, non-randomized, longitudinal, observational, prospective cohort study initially planned to follow participants for 48 weeks after an initial two-dose COVID-19 vaccine series with an optional study extension continued as a 96-week follow-up.

### 2.1. Recruitment

Following permission from the Ontario Ministry of Health, email introductions to the study were sent to persons who consented to be contacted for research at the time of COVID-19 vaccine administration. Similar email introductions were sent to members of the Ontario Canadian Association of retired persons (www.carp.ca, accessed on 20 January 2023) to increase capture of the older cohort. After confirmation of eligibility, participants enrolled through the study website, completed online health and demographic questionnaires and collected and submitted dried blood spots (DBSs) for antibody analysis [38]. We herein report results up to 96 weeks post the second vaccine dose (index date).

### 2.2. Inclusion Criteria

Two cohorts were enrolled: those between 30 and 50 years and those aged over 70 years, with planned over-representation of the older cohort. They were required to have received COVID-19 vaccine through an Ontario Distribution center. The only other exclusion criterion was an inability to understand study tasks and complete study documents in English or an inability to use an electronic device.

### 2.3. Consent

Agreement to participate in the initial and extended study, including authorization to share core data elements with the Canadian Immunity Task Force (funding agency), was obtained through e-consent. The University Health Network (UHN) Ethics Review Board Committee approved all study and consent processes and documents. Participants who e-consented were provided a personal identification (ID) number and created a password to enable communication with study staff, data collection and results reporting through their individual portal on the study website. Written and electronic schedules for required study activities were provided along with regular email reminders.

### 2.4. Study Population

The flow chart for the cohort is displayed in Figure 1. Between May 17 and July 31, 2021, 1286 persons (911 in the older cohort and 375 in the younger cohort) self-recruited to the main study. We excluded the following from this analysis: 2 participants who did not meet eligibility criteria, 79 who withdrew consent prior to the completion of any study activities and 1 who did not provide baseline demographic data, leaving 1204 (93.7%) continuing in the cohort. For this analysis, we defined the index date as the date of receipt of the second vaccine dose of the initial two-dose COVID-19 vaccine series. We included the 1022 participants who submitted at least one DBS after the index date and had not reported or had evidence of a COVID-19 infection before the index date.

For the analysis of the first breakthrough COVID-19 infection, we defined cases as participants who had at least one vaccine antibody value available at least one month after the second vaccine dose. A SARS-CoV-2 (COVID-19) infection was determined either by self-report of a positive polymerase chain reaction (PCR) or rapid antigen test (RAT) for COVID-19 or a positive NP (nucleoprotein) antibody level above the threshold on laboratory testing. The infection date was recorded as the first positive of either of the above. We excluded those reporting a possible COVID-19 infection on the basis of symptoms only. For the assessment of hybrid immunity (natural plus vaccine immunity), we determined a second breakthrough infection as the self-report of a new positive polymerase chain reaction (PCR) or rapid antigen test (RAT) for COVID-19 or a second positive NP (nucleoprotein) antibody level increasing above threshold (having decreased below that level after the first infection) on testing after 26 September 2022. Participants without a breakthrough infection during the 96-week follow-up required at least one antibody value available one month after the second vaccine dose to be included.

### 2.5. Outcome of Breakthrough Infection

For the analysis of SARS-CoV-2 breakthrough infections, the pre-identified predictors were assessed separately during the COVID-19 variant waves. The time intervals for the waves were allocated according to the peak infection periods in Ontario as indicated in Table 1.

### 2.6. Exposures

We included the participant’s baseline characteristics, vaccination status and RBD (receptor binding domain) IgG antibody levels in BAU/mL derived from the dried blood spots (DBSs).

### 2.7. Demographic Characteristics and Questionnaires

Baseline demographic, racial background and comorbidity health data were recorded by participants in their portal on the study website. They recorded the initial vaccine series administration dates and brands. Monthly email prompts reminded participants to complete online updates of any COVID-19 vaccine booster doses and dates with a dropdown list to indicate the brands. They were also asked to report any interval COVID-19 infections, date of onset and need for medical attention or hospitalization. Data were collected using the REDCap electronic data capture tool (version 13.1.28-© 2023 Vanderbilt University) hosted at the University Health Network and transferred and recorded in a specifically designed administrative section of the study; website www.stopcov.ca, accessed on 20 December 2023.

### 2.8. Bivalent Vaccination

In Canada, Spivax bivalent Original/Omicron BA.1 (Moderna, Laval, QC, Canada) was authorized on 26 September 2022, Comirnaty Original/Omicron BA.4/5 (Pfizer-BioNTech, Mainz, Germany) on 7 October 2022 and Spikevax bivalent Original/BA.4/5 (Moderna-NIAID, Rockville, MD, USA) on 3 November 2022. We defined bivalent booster vaccination as vaccination after 26 September 2022. The whole virion-inactivated COVID-19 vaccines (Spikevax XBB.1.5) were not available during the study period.

### 2.9. Sample Collection for Antibody Testing

Instructions for self-collection of DBS (by finger prick on Whatman 903 cards) and methods used to return specimens to the research unit were available in writing and in an online website video. Study staff were available for assistance by telephone or email. All necessary supplies and prepaid postage return envelopes were provided. Specimens were solicited two weeks after the second vaccine dose and then every 12 weeks within a window of ±3 weeks. In addition, we requested samples 3–4 weeks after the first two vaccine boosters [39].

### 2.10. Serological Assays, Interpretation and Recording

The DBS cards were checked for quality upon receipt, the dates and barcodes were recorded in the RED Cap database and samples were then transferred to the Lunenfeld-Tanenbaum Research Institute (LTRI) at Sinai Health for testing (Sinai REB study 21-0112-E). Previously validated enzyme-linked immunosorbent assays (ELISAs) for detecting antibodies (IgG) against the spike trimer, its receptor binding domain (RBD) and nucleocapsid proteins (NPs) [38,40] were employed at two sample dilutions. Luminescence values were normalized to a synthetic standard (relative ratios) for conversion to BAU/mL, which were then used to assign seropositivity thresholds (set at 99% specificity) as previously reported [37]. Monitoring anti-nucleocapsid antibodies enabled identification of possible new or re-infections (vaccines in Canada during the study period are based on spike antigen only).

### 2.11. Statistical Analysis

Baseline characteristics were stratified by the occurrence of a first breakthrough infection or not during follow-up and compared using chi-squared test, Fisher’s exact test or the Student’s t-test, as appropriate. The index date was defined as the date of the second vaccine dose and the follow-up period was up to 96 weeks after the index date. Cumulative incidence function curves were determined for the time to the first breakthrough infection stratified by age group. RBD (receptor binding domain) antibody values within one month of the second dose vaccine or one month prior to a breakthrough were excluded.

To examine the association between the risk of breakthrough infection and predictor variables, Cox proportional hazard regression models with time-dependent covariates were fitted. In order to investigate the time evolution of breakthrough infection, two different analyses were performed; see Table 2.

The first analysis (Model 1) used the time of first breakthrough infection from the index date (which occurred between May 26 and 16 September 2021) as the outcome, while the second analysis (Model 2) used the time of first breakthrough infection after 26 September 2022 as the outcome. The models also differed in terms of the risk set and the exposures. In Model 1, participants were followed from the index date until either their first breakthrough infection, three weeks after their final RBD collection date or 96 weeks—whichever occurred first. The latter two scenarios were considered as censoring events. In Model 2, participants were followed from 26 September 2022 until either a breakthrough infection, three weeks after their final RBD collection date after 26 September 2022 or 96 weeks after the index date—whichever occurred first. The latter two scenarios were considered as censoring events. By design, the risk set at time zero for Model 1 contained all 983 participants who received their initial two doses of any SARS-CoV-2 vaccine; in contrast, the risk set at time zero for Model 2 was restricted to the 776 participants who had any follow-up data available after 26 September 2022. For both models, time-dependent covariates selected a priori included log-transformed RBD antibody levels. Given the strong correlation between the number of vaccine doses and the antibody levels, the latter could not be included in the models. For both models, fixed covariates included age group (30–50 years, 70+ years), sex (male, female), Caucasian ethnicity, comorbidity (presence of diabetes, respiratory disease, cardiovascular disease, transplant/immunosuppression or cancer), current smoking status and initial two-dose vaccine brand (two Moderna (mRNA-1273) vs. other). However, in Model 2, we included a confirmed history of COVID-19 infection prior to 26 September 2022 and receipt of an original/Omicron booster after 26 September 2022 as additional fixed covariates. Univariable and multivariable models were constructed, and tests of the proportional assumptions were performed. An α-level of 0.05 was used for statistical hypothesis tests. Analyses were performed using R software (4.2.2) and the survival package.

For Model 1, a sensitivity analysis was performed that considered first breakthrough infection occurring in either the Delta + Omicron BA.1/2 waves or the Omicron BA.4/5 + XBB waves. For the former, participants who did not experience a breakthrough infection were censored at the end of the wave. For the latter, participants who experienced an infection or were censored prior to the start of the wave were removed from the risk set.

### 2.12. Data Access

As part of a standard sharing agreement with the Public Health Agency of Canada, the relevant anonymized data together with a dictionary defining each variable have been transferred to the Canadian COVID-19 Immunity Task Force (CITF) and stored at McGill University. Access by any external researchers requires submission of a request evaluated by a CITF checklist to ensure conformity to the privacy and ethical protocols and compliance with Canadian law and research ethics, which is then forwarded for consideration for approval by the Data Access Committee.

## 3. Results

Overall, 1286 participants signed the e-consent for the original study, 2 did not meet the eligibility criteria, 1 did not submit baseline data and 79 failed to complete any study activities. Among 1204 eligible participants, the following were excluded from this analysis: 72 participants who reported that they suspected that they had COVID-19 infection before receiving a second dose of vaccine, an additional 17 participants who did not report a COVID-19 infection but showed a positive anti-nucleocapsid antibody indicating natural infection prior to the second vaccine dose and 20 participants who did not complete the initial two vaccine series. Of the remaining, 1022 participants had at least one antibody level available. Nine hundred and eighty-three had at least one antibody value between one month after the second dose and one month before the breakthrough infection or at least one antibody value one month after the second vaccine dose. The final cohort for this analysis is 983 participants as illustrated in Figure 1.

The demographics and vaccination status of the population in this analysis are displayed in Table 3. Of 983 participants, 765 (74.8%) were older than 70 years old, 631 were (64.2%) female, 872 (88.7%) Caucasian, 38 (3.9%) current smokers, 237 (24.1%) classified as obese (body mass index ≥ 30), 106 (10.8%) had diabetes, 115 (11.7%) respiratory diseases, 159 (16.2%) cancer, 357 (36.3%) cardiovascular disease and 42 (4.3%) with transplant or immunosuppression.

Overall, 310 (31.5%) participants received at least one mRNA-1273 (Moderna) vaccine dose in the initial two-dose series. The number and timing of vaccine boosters was guided by the public health recommendations. For the younger cohort, n = 30 (12%) participants, 81 (33%), 90 (36%), 41 (16.5%) and 6 (2.4%) received 0, 1, 2, 3 or 4 booster doses, respectively, over the 96-week study period. The corresponding numbers for the older cohort are 37 (5%), 96 (13%), 143 (19.5%), 377 (51.2%,) and 82 (11%). In total, 318 (32%) had received one original/Omicron booster, including 37 (15%) of the younger and 281 (38%) of the older cohort.

In total, 465 (47.3%) first breakthrough infections were observed within 96 weeks of the index date, 376 (76%) were reported by participants (of these, 206 (55%) also had a positive NP) and an additional 89 (19%) had a positive NP but did not report a positive PCR or RAT. As shown in Figure 2, the younger-aged cohort was more likely (*p*-value < 0.0001) to have a breakthrough infection with a median time of 425 [range 395, 487] days compared to a median time to infection of 669 days in the older cohort [range 638, >672]. A total of 289 (62%) of the first breakthrough events occurred before 26 September 2022, and 176 (38%) occurred after this date.

The number of breakthrough infections increased with time. A total of 20 (4.3%) breakthrough COVID-19 infections were observed during the Delta wave with a median time of 122 days (17 weeks) from the index date to infection date; 53 (11.4%) during the Omicron BA.1 wave with a median time of 214 days (31 weeks); 127 (27.3%) during the Omicron BA.2 wave with a median time of 304 days (43 weeks); 238 (51.2%) during the Omicron BA.4/5 wave with a median time of 487 days (70 weeks); and 27 (5.8%) during the Omicron XBB wave with a median time of 669 days (96 weeks). Of the 776 participants with follow-up data, 221 (28.5%) had a breakthrough infection after 26 September 2022 and 289 (37.2%) had a previous documented infection prior to 26 September 2022. Of the latter, 45 had a second breakthrough infection after 26 September 2022.

Overall, during the 96 weeks after the index date, higher RBD antibody levels were protectively associated with breakthrough infection, with a hazard ratio (HR) of 0.933 [95% CI 0.872, 0.999] and, similarly, those in the older-aged cohort of 70+ were protectively associated with an HR of 0.451 [95% CI, 0.358, 0.568], as shown in Table 4 and Figure 3a. Higher antibody values and the older-aged cohort in the earlier infection waves of Delta, Omicron BA.1 and Omicron BA.2 had a stronger protective association with breakthrough infection, respectively, with HRs of 0.884 [95% CI, 0.813, 0.963] and 0.299 [95% CI, 0.211, 0.424]. However, in later waves of Omicron BA.4/5 and XBB, higher antibody levels were not associated with breakthrough infection, with an HR of 1.01 [95% CI, 0.92, 1.109], while the older-aged cohort remained associated with protection, with an HR of 0.512 [95% CI, 0.373, 0.702]; see Table 5.

Receipt of two doses of the mRNA-1273 vaccine in the initial series was not associated with protection from breakthrough infection during the 96-week follow-up—HR 0.862 [95% CI 0.624, 1.192]—or during any SARS-CoV-2 waves. The covariates of sex, race, smoking habit and comorbidities (diabetes, respiratory disease, cardiovascular disease, transplant/immunosuppression and cancer) were not associated with breakthrough infection during the 96 weeks of follow-up or during any infection wave; see Table 5.

The higher RBD antibody quintiles were associated with the first breakthrough infection during the earlier waves of Delta and Omicron BA.1 in Ontario; HR 0.302 [95% CI, 0.117, 0.776] and HR 0.205 [95% CI, 0.082, 0.508], as shown in Table 6. The older-aged cohort (70+ years) was protectively associated with breakthrough infection during all waves in Ontario (Table 7).

For those with a single COVID-19 breakthrough infection, the median time to breakthrough infection after the second vaccine dose was 426 days (61 weeks). This group received an average of 1.7 booster doses before the breakthrough. For those with a second breakthrough infection, the median time to the second infection after the first infection was 248 days (35 weeks), with receipt of an average of 0.9 booster vaccine doses between the two infections.

The potential association of receipt of the Omicron original/Omicron vaccine booster after 26 September 2022 and the presence of natural hybrid infection before 26 September 2022 on the risk of second infections was investigated. There was no association of higher RBD antibody levels—HR 1.045 [95% CI, 0.94, 1.163]—and initial vaccine series with two doses of mRNA-1273—HR 0.773 [95% CI, 0.467, 1.28]—however, the older age cohort was protectively associated; HR 0.682 [95% CI, 0.477, 0.976]. Both the receipt of a bivalent original/Omicron booster vaccine—HR 0.36 [95% CI, 0.264, 0.49]—and previous natural infection (hybrid immunity)—HR 0.276 [95% CI 0.189, 0.404]—were protectively associated with breakthrough infection after 26 September 2022, as shown in Table 8, and Figure 3b.

## 4. Discussion

COVID-19 mRNA vaccines have proven to be very effective in preventing severe infection and death but do not completely prevent SARS-CoV-2 infection [19,21,22,24,41]. In the STOPCoV prospective study [37], the proportion of participants who developed natural infection increased over the study period, especially during the Omicron SARS-CoV-2 variant waves, despite high rates of vaccine and booster uptake. No participant required hospitalization or died consequent to their infection.

In this analysis, we explored predictors of breakthrough infection in the STOPCoV study. Our main goal was to see if we could identify an antibody threshold for protection. In the early phases of the pandemic after the initial release of vaccines, we found an association between the level of antibodies in the spike receptor binding domain (RBD) of the SARS-CoV-2 virus and a lower rate of breakthrough infection. However, with time and the emergence of new SARS-CoV-2 variants, the association between the RBD antibody level and breakthrough infection became weaker and lost statistical significance. We similarly could not identify a threshold level that provided protection when analyzing breakthrough infection using RBD antibody quintiles. Thus, within our cohort, RBD antibody levels could not be used to predict the risk of infection or guide booster dosing.

To date, there is no confirmed absolute antibody threshold for immunity. Antibody positivity was a good predictor of lower infection risk in a health care worker study, but a comparison of antibody levels was not carried out [42]. A 2310-participant Israelian health care worker cohort showed a lower risk of infection for each 10-fold increase in pre-infection IgG antibody levels to spike (OR 0.71, 95% CI 0.56–0.90), with a higher risk of symptomatic disease with pre-infection IgG levels < 500 BAU/mL compared to those with levels 901–1600 BAU/mL [43]. The TWINS UK and ALSPAC UK longitudinal study of 9361 individuals determined that those with the lowest quintile antibody to spike (< 164 BAU/mL) after a single vaccine had greater odds of a COVID-19 infection in the next 6–9 months (OR 2.9, 95% CI 1.4, 6.0) compared to those in the top antibody quintile [15]. Protective thresholds may vary with the SARS-CoV-2 variant. For example, Goldblat et al. [44] defined 154 (95% CI 42, 559) and 171 (95% CI 57, 519) BAU/mL as protective thresholds for wild-type and alpha variant SARS-CoV-2 and 39–490 BAU/mL for delta variants while Feng et al. [18] found levels above 264 (95% CI 108, 806) BAU/mL to provide 80% vaccine effectiveness against the Alpha variant. Dimeglio et al. [45] estimated much higher levels > 6000 BAU/mL needed for protection against Omicron variant BA.1 but could not define an antibody threshold for Omicron variant BA.2. Similar to our work, an Italian study of 487 individuals [17] showed no significant difference in the probability of SARS-CoV-2 infection for different levels of anti-spike/RBD from values obtained one month after the first COVID-19 booster and 4 months of follow-up. They recommend against routine testing for vaccine-induced humoral immune responses, and our data would support that recommendation.

Others have suggested that vaccine-induced neutralizing antibodies may have a greater predictive power against severe infection [11,19]. Neutralizing antibodies induced by a vaccine were shown to strongly correlate with the protective effect of seven COVID vaccines in the phase III clinical trials [11,46]. However, the gradient of vaccine efficacy increases with neutralization, with 70% protective thresholds ranging from 4 to 33/U/mL but with no strict threshold confirmed [11,18,47]. Although the studies show similar correlations, the results are difficult to compare across studies because of the diversity of the assays used, neutralization levels between assays, the different populations studied, the duration of follow-up, the number and types of vaccines received and the constant emergence of new and more escaped variants [16].

We did not find the rate of breakthrough infections to vary by race, gender or in those with underlying comorbidities or a smoking status. Breakthrough infection rates, particularly in the early waves of infection, are reported as higher in those with underlying immunosuppressive disease or therapy [6,48,49,50,51,52,53]. The number of participants with these disorders in our cohort was small, potentially preventing us from identifying a relationship. However, overall, the proportion with a breakthrough infection did not differ between those who did and did not report an underlying comorbidity. We did not find an association between breakthrough infection and the receipt of two doses of mRNA-1273 as the initial vaccine series, although other reports have seen higher antibody levels with this initial brand as the vaccine series [54,55,56].

In our study, at all time points, and throughout all variant waves of infection in Ontario, our older cohort (≥70 years) was less likely to have a breakthrough infection than the younger cohort (30–50 years). This was consistent in the multivariable analysis when considering RBD antibody levels. Others have similarly described a lower rate in the older population [23,57]. Although one might anticipate that the older cohort would have a lower immune response, this was not reflected in lower antibody levels. We speculate that lower breakthrough infection rates may be related to behavioral differences and better adherence to public health recommendations concerning social distancing and isolation in a group who perceived themselves of higher risk and have more underlying comorbidities; however, these data were not prospectively collected.

Multiple studies have shown waning vaccine efficacy with time and against the emerging variants. Vaccine efficacy may be partially restored with vaccine booster doses but wanes rapidly [29,58,59,60]. Although repeated boosting with the earliest vaccines based on the ancestral SARS-CoV-2 strain increased antibody levels, an Israeli study suggested that, after three doses, the immunogenicity peaks and that the fourth dose only offers marginal protection in young adults [48]. The antibody levels in our cohort were high and we could only show an association with breakthrough infection in the early waves of the pandemic. Similar to others, the protective threshold (if it exists) may vary with the circulating variant strain [30].

The emergence of breakthrough infections has led to the development of boosters with activity against the circulating SARS-CoV-2 variants [26,35,61,62]. Laboratory studies have shown that antibodies from persons boosted with original/Omicron vaccines had an increased neutralization capacity [60,63]. Multiple test-negative case–control studies in the US, Japan, Korea, Israel and Nordic countries have shown decreased rates of hospitalization and death due to COVID-19 in those who received an original/Omicron vaccine relative to those who did not, with additional vaccine effectiveness ranging from 45 to 89%. The studies include different populations and different prior vaccine schedules [25,27,31,35,36,64,65,66,67,68].

When we assessed new breakthrough infections after the release of the original/Omicron vaccine in Ontario, we found a strong association between receipt of the bivalent booster and lack of breakthrough infection. This suggests that having a vaccine that more closely matches the circulating strain may provide the greatest protection and points to the likely future need to have updated boosters on an annual basis that match the new variants similar to the practice with influenza vaccination. These observations further call for the need for a pan-coronavirus vaccine. The impact of the new monovalent vaccines on the emerging SARS-CoV-2 variants will be an area of ongoing study.

We also demonstrated that hybrid immunity—the presence of both natural and vaccine-induced immunity—was associated with a lower rate of subsequent breakthrough infection after the original/Omicron vaccines were introduced in Ontario. This is consistent with other studies that demonstrate the partial protection afforded by natural infection [7,8,69,70,71].

Our study has many strengths. The completely decentralized nature of our study allowed for the inclusion of a more generalizable population including persons outside of traditional research facilities or with limitations in participation due to mobility. Further, the ability to complete study procedures at home enabled high retention and follow-up [39]. The population studied had a high vaccine and booster uptake. We were able to follow the cohort longitudinally and with high retention, allowing us to assess the predictors through multiple waves of the pandemic. Despite this, the number with breakthrough COVID-19 infections remains small relative to population-based studies and there was an uneven representation of the various SARS-CoV-2 variant strains causing breakthroughs over the course of our study. We also had less power to assess for many of the comorbidity covariates. We relied on participant reporting of infection and vaccine brands and booster doses, which could lead to some misclassification. In those cases that were minimally symptomatic or asymptomatic, the true date of infection was unclear as the antibody to NP was only determined every three months. As the reporting of infection was monthly, antibody values within the month of the breakthrough were not included in the analysis as the levels may be enhanced by the infection. Lastly, the chosen NP threshold of 99% specificity as an indication of infection for the study may have misclassified cases in which antibody production was lower as only 55% of those reporting an infection showed a subsequent increase in NP. It is possible that the production of NP is lower in a vaccinated population or that it could have waned between testing periods. There was a small percentage (<5%) of antibody levels that were above the linear range of the assay but we do not feel that they are large enough to prevent us from determining an antibody threshold if it existed. Finally, antibody tests were tailored to the ancestral strains and, in our future work, will determine whether they need to be adapted to the new SARS CoV-2 variants. Our study only considered antibody levels, and neutralizing antibodies and T cell immunity were not assessed, which could be impacted by age and underlying risk factors.

## 5. Conclusions

This prospective longitudinal study failed to identify an antibody threshold that would protect against infection with SARS-CoV-2. We conclude that antibody levels based on the ancestral strain cannot be used to determine the need for or timing of booster dosing. Although there was an association between higher antibody levels and a lack of breakthrough infection in the early waves of the pandemic, this decreased with time and with the emergence of SARS-CoV-2 variant strains. There was an association between receipt of an original/Omicron vaccine and lack of a breakthrough infection during the Omicron BA.4/5 waves of the pandemic, suggesting that receipt of vaccines that match the circulating variants may provide the greatest protection and supports the recommendation on their use in an era of vaccine fatigue. Hybrid immunity was also associated with protection from further infections.

## Figures and Tables

**Figure 1 vaccines-12-00036-f001:**
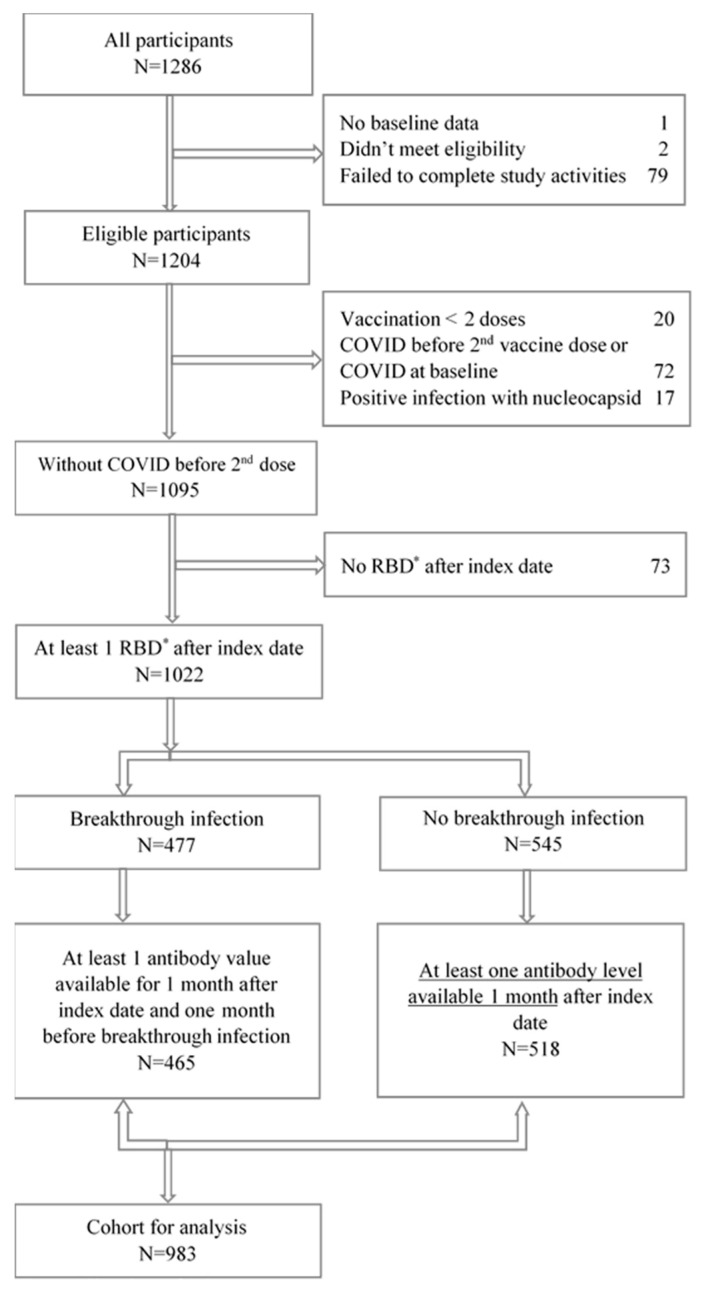
Flowchart of Cohort. * RBD: Receptor Binding Domain.

**Figure 2 vaccines-12-00036-f002:**
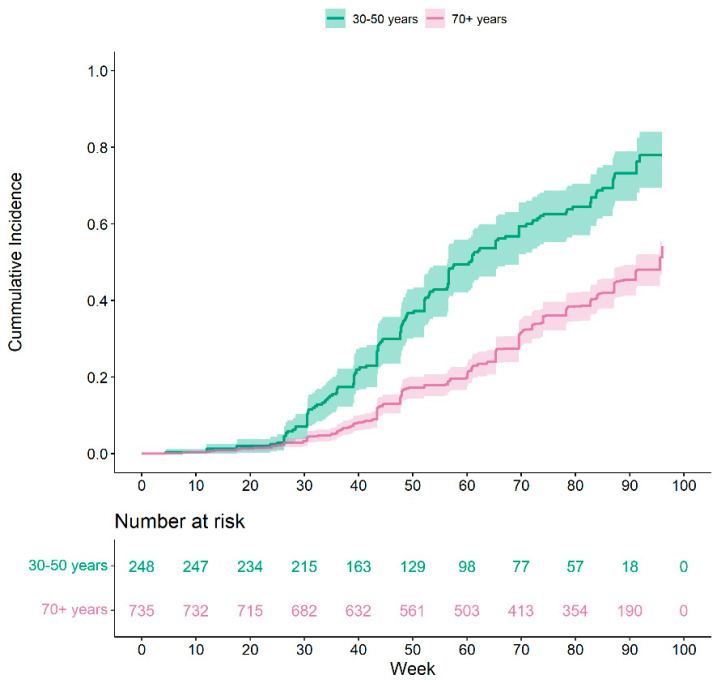
Cumulative incidence function of COVID-19 breakthrough infection stratified by age cohort.

**Figure 3 vaccines-12-00036-f003:**
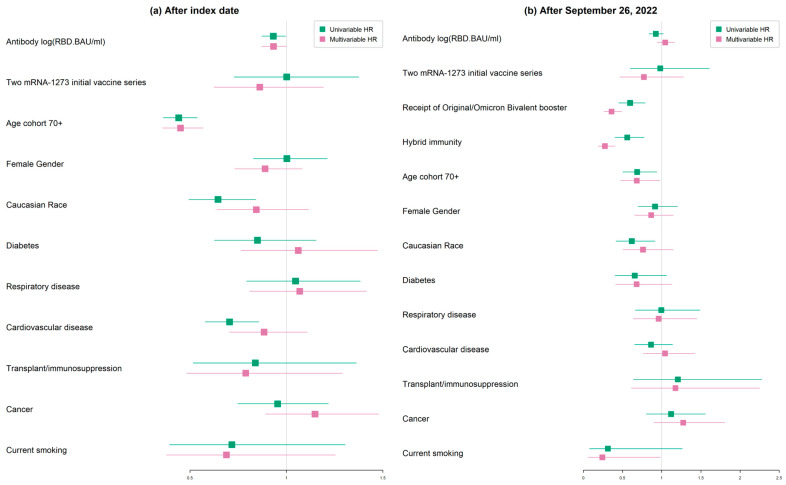
Hazard ratios of predictors in association with breakthrough COVID-19 infection (**a**) after index date and (**b**) after 26 September 2022. RBD, receptor binding domain; BAU, binding antibody units/mL.

**Table 1 vaccines-12-00036-t001:** Peak COVID-19 infection variant wave periods in Ontario.

COVID-19 Variant Infection Wave	Calendar Time Interval
Delta	1 August 2021–14 December 2021
Omicron BA.1	15 December 2021–28 February 2022
Omicron BA.2	22 March 2022–18 June 2022
Omicron BA.4/5	19 June 2022–25 March 2023
Omicron XBB	26 March 2023–present

**Table 2 vaccines-12-00036-t002:** Settings of Cox regression analyses for index date and 26 September 2022.

Components	Cox Regression after Index Date	Cox Regression after 26 September 2022
Outcome	The first breakthrough after index date during 96-week follow-up period	The first breakthrough infection after 26 September 2022 up to 96 weeks after the index date.
Follow-up date	Infected: the first breakthrough infection date during the 96-week follow-up after the index date. Non-infected: Three weeks after the last RBD date up to 96 weeks after the index date.	Infected: the first breakthrough infection date after 26 September 2022 up to 96 weeks after the index date. Non-infected: three weeks after the last RBD date up to 96 weeks after the index date.
Risk set	983 participants.	776 participants; participants with follow-up dates before 26 September 2022 are excluded.
Exposures	RBD values during the 96-week follow-up and the first two vaccine brands.	RBD values during the 96-week follow-up, the first two vaccine brands, receipt of bivalent booster after 26 September 2022 and hybrid infection before 26 September 2022.

RBD—receptor binding domain.

**Table 3 vaccines-12-00036-t003:** Comparison of baseline characteristics stratified by those with and without a breakthrough COVID-19 infection.

Characteristics	No Breakthrough SARS-CoV-2 Infection	Breakthrough SARS-CoV-2 Infection	*p*-Value
	n = 518	n = 465	
Age group			
30–50-year cohort	98 (18.9%)	150 (32.3%)	<0.001
70+ year cohort	420 (81.1%)	315 (67.7%)	
Age, mean (SD)	67.7 (14.2)	63.5 (16.2)	<0.001
Sex			
Male	186 (35.9%)	166 (35.7%)	0.999
Female	332 (64.1%)	299 (64.3%)	
Caucasian Race			
No	48 (9.27%)	63 (13.5%)	0.044
Yes	470 (90.7%)	402 (86.5%)	
Diabetes			
No	458 (88.4%)	419 (90.1%)	0.453
Yes	60 (11.6%)	46 (9.89%)	
Cardiovascular diseases			
No	307 (59.3%)	319 (68.6%)	0.003
Yes	211 (40.7%)	146 (31.4%)	
Respiratory diseases			
No	460 (88.8%)	408 (87.7%)	0.676
Yes	58 (11.2%)	57 (12.3%)	
Cancer			
No	437 (84.4%)	387 (83.2%)	0.692
Yes	81 (15.6%)	78 (16.8%)	
Transplant			
No	493 (95.2%)	448 (96.3%)	0.455
Yes	25 (4.83%)	17 (3.66%)	
Body mass index, mean (SD)	2.71 (0.55)	2.71 (0.55)	0.941
Obesity (BMI ≥ 30)			
No	381 (74.6%)	353 (76.7%)	0.475
Yes	130 (25.4%)	107 (23.3%)	
Smoking			
Never	282 (54.4%)	277 (59.6%)	0.037
Prior	209 (40.3%)	177 (38.1%)	
Current	27 (5.21%)	11 (2.37%)	
First two vaccine brand combination			
mRNA-1273–mRNA-1273	59 (11.4%)	42 (9%)	0.535
Other–mRNA-1273	104 (20.1%)	105 (22.6%)	
Pfizer–Pfizer	329 (63.5%)	297 (63.9%)	
Other–Other	26 (5%)	21 (4.5%)	
COVID-19 Vaccine Booster doses			
None	47 (9.07%)	20 (4.3%)	<0.001
One	111 (21.43%)	66 (14.19%)	
Two	92 (17.76%)	141 (30.32%)	
Three	215 (41.51%)	203 (43.66%)	
Four	53 (10.23%)	35 (7.53%)	
Original/Omicron Bivalent vaccination (after 26 September 2022)			
No	276 (53.3%)	389 (83.7%)	<0.001
Yes	242 (46.7%)	76 (16.3%)	

BMI—body mass index, SD—standard deviation.

**Table 4 vaccines-12-00036-t004:** Hazard ratios of predictors in association with COVID-19 breakthrough infection after index date.

Characteristics	Univariable Hazard Ratio [95% CI]	Multivariable Hazard Ratio [95% CI]
96 weeks after index date		
Antibody log (RBD.BAU/mL)	0.932 [0.873, 0.996]	0.933 [0.872, 0.999]
Two mRNA-1273 initial vaccine series	1.001 [0.729, 1.375]	0.862 [0.624, 1.192]
Age cohort 70+	0.442 [0.363, 0.538]	0.451 [0.358, 0.568]
Female gender	1.002 [0.829, 1.211]	0.889 [0.731, 1.081]
Caucasian race	0.645 [0.494, 0.841]	0.844 [0.639, 1.115]
Diabetes	0.85 [0.627, 1.153]	1.061 [0.765, 1.471]
Respiratory disease	1.047 [0.794, 1.382]	1.069 [0.808, 1.415]
Cardiovascular disease	0.705 [0.579, 0.857]	0.884 [0.705, 1.108]
Transplant/immunosuppression	0.839 [0.517, 1.362]	0.789 [0.482, 1.29]
Cancer	0.954 [0.748, 1.217]	1.148 [0.891, 1.478]
Currently smoking	0.717 [0.394, 1.304]	0.689 [0.378, 1.254]
Delta + Omicron BA.1/.2 waves		
Antibody log (RBD.BAU/mL)	0.899 [0.829, 0.975]	0.884 [0.813, 0.963]
Two mRNA-1273 initial vaccine series	1.244 [0.799, 1.939]	1.057 [0.669, 1.67]
Age 70+ cohort	0.348 [0.261, 0.463]	0.299 [0.211, 0.424]
Female gender	0.967 [0.724, 1.291]	0.799 [0.59, 1.081]
Caucasian race	0.785 [0.516, 1.194]	1.227 [0.785, 1.918]
Diabetes	0.966 [0.614, 1.518]	1.267 [0.777, 2.065]
Respiratory disease	1.096 [0.721, 1.667]	1.047 [0.68, 1.611]
Cardiovascular disease	0.694 [0.514, 0.936]	0.96 [0.672, 1.371]
Transplant/immunosuppression	0.926 [0.436, 1.97]	0.836 [0.389, 1.798]
Cancer	1.013 [0.701, 1.466]	1.312 [0.888, 1.938]
Currently Smoking	1.325 [0.653, 2.688]	1.252 [0.612, 2.563]
Omicron BA.4/5 + XBB waves		
Antibody log (RBD.BAU/mL)	0.99 [0.904, 1.084]	1.01 [0.92, 1.109]
Two mRNA-1273 initial vaccine series	0.857 [0.543, 1.351]	0.692 [0.435, 1.101]
Age 70+ cohort	0.464 [0.353, 0.61]	0.512 [0.373, 0.702]
Female gender	1.02 [0.793, 1.312]	0.9 [0.695, 1.165]
Caucasian race	0.526 [0.373, 0.742]	0.645 [0.45, 0.926]
Diabetes	0.77 [0.51, 1.163]	0.951 [0.611, 1.48]
Respiratory disease	1.023 [0.707, 1.48]	1.054 [0.725, 1.533]
Cardiovascular disease	0.695 [0.537, 0.901]	0.828 [0.616, 1.111]
Transplant/immunosuppression	0.832 [0.442, 1.565]	0.835 [0.44, 1.584]
Cancer	0.899 [0.65, 1.242]	1.055 [0.754, 1.476]
Currently smoking	0.338 [0.108, 1.056]	0.321 [0.103, 1.004]

RBD—receptor binding domain, BAU—binding antibody assay.

**Table 5 vaccines-12-00036-t005:** Baseline characteristics of participants with breakthrough COVID-19 infection stratified by Ontario SARS-CoV-2 waves.

Characteristics	COVID-19 Breakthrough Infected Participants					*p*-Value
	Delta (n = 20)	BA.1 (n = 53)	BA.2 (n = 127)	BA.4/5 (n = 238)	XBB (n = 27)	
Age cohort						
30–50 years	5 (25.0%)	29 (54.7%)	45 (35.4%)	66 (27.7%)	5 (18.5%)	<0.001
70+ years	15 (75.0%)	24 (45.3%)	82 (64.6%)	172 (72.3%)	22 (81.5%)	
Age, mean (SD)	64.4 (18.4)	57.1 (18.4)	61.8 (16.4)	65.1 (15.5)	68.7 (11.4)	0.005
Sex						
Male	8 (40.0%)	21 (39.6%)	43 (33.9%)	83 (34.9%)	11 (40.7%)	0.898
Female	12 (60.0%)	32 (60.4%)	84 (66.1%)	155 (65.1%)	16 (59.3%)	
Caucasian Race						
No	2 (10.0%)	6 (11.3%)	17 (13.4%)	36 (15.1%)	2 (7.41%)	0.86
Yes	18 (90.0%)	47 (88.7%)	110 (86.6%)	202 (84.9%)	25 (92.6%)	
Diabetes						
No	19 (95.0%)	51 (96.2%)	109 (85.8%)	221 (92.9%)	19 (70.4%)	0.002
Yes	1 (5.00%)	2 (3.77%)	18 (14.2%)	17 (7.14%)	8 (29.6%)	
Cardiovascular disease						
No	19 (95.0%)	40 (75.5%)	79 (62.2%)	165 (69.3%)	16 (59.3%)	0.024
Yes	1 (5.00%)	13 (24.5%)	48 (37.8%)	73 (30.7%)	11 (40.7%)	
Respiratory diseases						
No	17 (85.0%)	47 (88.7%)	111 (87.4%)	209 (87.8%)	24 (88.9%)	0.991
Yes	3 (15.0%)	6 (11.3%)	16 (12.6%)	29 (12.2%)	3 (11.1%)	
Cancer						
No	17 (85.0%)	43 (81.1%)	106 (83.5%)	199 (83.6%)	22 (81.5%)	0.984
Yes	3 (15.0%)	10 (18.9%)	21 (16.5%)	39 (16.4%)	5 (18.5%)	
Transplant/immunosuppression						
No	19 (95.0%)	50 (94.3%)	124 (97.6%)	229 (96.2%)	26 (96.3%)	0.646
Yes	1 (5.00%)	3 (5.66%)	3 (2.36%)	9 (3.78%)	1 (3.70%)	
Obesity						
No	13 (68.4%)	38 (73.1%)	93 (74.4%)	189 (79.7%)	20 (74.1%)	0.527
Yes	6 (31.6%)	14 (26.9%)	32 (25.6%)	48 (20.3%)	7 (25.9%)	
BMI, mean (SD)	2.71 (0.65)	2.80 (0.57)	2.78 (0.59)	2.65 (0.53)	2.77 (0.44)	0.171
Smoking						
No	8 (40.0%)	30 (56.6%)	77 (60.6%)	150 (63.0%)	12 (44.4%)	0.002
Previous	11 (55.0%)	17 (32.1%)	49 (38.6%)	86 (36.1%)	14 (51.9%)	
Current	1 (5.00%)	6 (11.3%)	1 (0.79%)	2 (0.84%)	1 (3.70%)	
First two vaccine combination						
Other–Other	15 (75.0%)	33 (62.3%)	86 (67.7%)	167 (70.2%)	17 (63.0%)	0.453
Other–mRNA-1273	5 (25.0%)	15 (28.3%)	24 (18.9%)	53 (22.3%)	8 (29.6%)	
mRNA-1273–mRNA-1273	0 (0.00%)	5 (9.43%)	17 (13.4%)	18 (7.56%)	2 (7.41%)	
Original/Omicron Bivalent vaccination (after 26 September 2022)						
No	20 (100%)	53 (100%)	127 (100%)	178 (74.8%)	11 (40.7%)	<0.001
Yes	0 (0.00%)	0 (0.00%)	0 (0.00%)	60 (25.2%)	16 (59.3%)	
Median time to infection, days (week)	122 (17)	214 (31)	304 (43)	487 (70)	669 (96)	-

RBD—receptor binding domain, BAU—binding antibody unit, CI—confidence interval.

**Table 6 vaccines-12-00036-t006:** Hazard ratios (HRs) of quintiles to receptor binding domain (RBD) antibodies in association with breakthrough infection after the index date.

Antibody Quintiles of Log (RBD.BAU/mL)	Univariable HR [95% CI]	Multivariable HR [95% CI]
All SARS-CoV-2 waves		
<4.92	Ref.	
4.92–6.03	0.851 [0.576, 1.256]	0.823 [0.557, 1.216]
6.03–7.13	0.795 [0.549, 1.152]	0.77 [0.53, 1.118]
7.13–8.13	0.861 [0.602, 1.233]	0.872 [0.608, 1.252]
>8.13	0.688 [0.475, 0.996]	0.689 [0.474, 1]
Delta + Omicron BA.1		
<4.92	Ref.	
4.92–6.03	0.974 [0.521, 1.818]	0.885 [0.47, 1.666]
6.03–7.13	0.955 [0.487, 1.875]	0.88 [0.443, 1.748]
7.13–8.13	0.292 [0.114, 0.747]	0.302 [0.117, 0.776]
>8.13	0.198 [0.08, 0.486]	0.205 [0.082, 0.508]
Omicron BA.2		
<4.92	Ref.	
4.92–6.03	0.846 [0.35, 2.044]	0.674 [0.276, 1.646]
6.03–7.13	0.755 [0.332, 1.718]	0.645 [0.282, 1.476]
7.13–8.13	0.96 [0.448, 2.058]	0.746 [0.344, 1.616]
>8.13	1.192 [0.563, 2.523]	0.894 [0.417, 1.918]
Omicron BA.4/5		
<4.92	Ref.	
4.92–6.03	1.243 [0.615, 2.511]	1.226 [0.603, 2.489]
6.03–7.13	1.291 [0.661, 2.522]	1.256 [0.638, 2.472]
7.13–8.13	1.508 [0.782, 2.908]	1.562 [0.805, 3.033]
>8.13	0.977 [0.495, 1.929]	1.012 [0.509, 2.012]
XBB		
<0.92	Ref.	
4.92–6.03	---	---
6.03–7.13	0.349 [0.102, 1.192]	0.414 [0.101, 1.7]
7.13–8.13	0.475 [0.151, 1.492]	0.618 [0.163, 2.35]
>8.13	0.404 [0.108, 1.503]	0.57 [0.132, 2.464]

RBD—receptor binding domain, BAU—binding antibody unit, CI—confidence interval.

**Table 7 vaccines-12-00036-t007:** Breakthrough infections by age cohort during the COVID waves.

Waves and Age Cohorts	Univariable HR [95% CI]	Multivariable HR [95% CI]
All SARS-CoV-2 waves		
30–50 years	Ref.	
70+ years	0.442 [0.363, 0.538]	0.448 [0.355, 0.564]
Delta + Omicron BA.1		
30–50 years	Ref.	
70+ years	0.277 [0.175, 0.439]	0.312 [0.179, 0.542]
Omicron BA.2		
30–50 years	Ref.	
70+ years	0.361 [0.251, 0.52]	0.326 [0.21, 0.506]
Omicron BA.4/5		
30–50 years	Ref.	
70+ years	0.464 [0.349, 0.617]	0.519 [0.372, 0.723]
Omicron XBB		
30–50 years	Ref.	
70+ years	0.361 [0.137, 0.955]	0.232 [0.074, 0.73]

HR—hazard ratio, CI—confidence interval.

**Table 8 vaccines-12-00036-t008:** Hazard ratios of predictors associated with first breakthrough COVID-19 infection after 26 September 2022.

Characteristic	Univariable HR [95% CI]	Multivariable HR [95% CI]
Antibody level log (RBD.BAU/mL)	0.925 [0.839, 1.02]	1.045 [0.94, 1.163]
Two mRNA-1273 doses in initial vaccine series	0.981 [0.598, 1.609]	0.773 [0.467, 1.28]
Receipt of original/Omicron bivalent booster	0.597 [0.452, 0.788]	0.36 [0.264, 0.49]
Hybrid immunity	0.56 [0.403, 0.777]	0.276 [0.189, 0.404]
Age cohort 70+	0.687 [0.504, 0.936]	0.682 [0.477, 0.976]
Female gender	0.915 [0.697, 1.202]	0.865 [0.653, 1.147]
Caucasian race	0.618 [0.418, 0.914]	0.761 [0.506, 1.145]
Diabetes	0.656 [0.405, 1.062]	0.68 [0.411, 1.128]
Respiratory disease	0.995 [0.665, 1.488]	0.96 [0.638, 1.446]
Cardiovascular disease	0.863 [0.654, 1.139]	1.043 [0.763, 1.424]
Transplant/immunosuppression	1.206 [0.64, 2.275]	1.175 [0.613, 2.25]
Cancer	1.119 [0.804, 1.557]	1.274 [0.901, 1.803]
Current smoking	0.314 [0.078, 1.263]	0.243 [0.06, 0.982]

HR—hazard ratio, RBD—receptor binding domain, BAU—binding antibody unit.

## Data Availability

Data available on request due to restrictions. The data presented in this study are available on request from the corresponding author. The data are not publicly available due to privacy. Data can be requested through Canadian Immunity Task Force.

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
