# Peer review of "Predictors of Breakthrough SARS-CoV-2 Infection after Vaccination"

_vaccines, 2023, doi:10.3390/vaccines12010036_

Round 1
Reviewer 1 Report
Comments and Suggestions for Authors
Estimated Authors of the paper "Predictors of Breakthrough SARS-COV-2 Infection after Vaccination",
I've read with great interest the present study, whose content deals with a significant Public Health topic, particularly when most of Western Countries struggle with 2023 COVID-19 vaccination campaigns. Interestingly, the present study reports on the breakthrough infections after bivalent COVID-19 vaccination, an intervention that has been recently replaced (at least in official recommendations) by improved monovalent vaccines, not only in the USA (https://www.cdc.gov/vaccines/hcp/acip-recs/vacc-specific/covid-19.html) but also in EU/EEA (https://www.ecdc.europa.eu/en/publications-data/interim-public-health-considerations-covid-19-vaccination-roll-out-during-2023#:~:text=to%20maximise%20individual%20protection%2C%20the,age%2C%20should%20also%20be%20considered).
The present study benefits from a very well defined design, and an accurate reporting, despite the total number of included cases in relatively small when compared to the cases of SARS-CoV-2 infections, particularly after the replacement of previous strains with Omicron VOC.
From my point of view, the content of this paper deserves a full publication, but several improvements could be suggested.
To begin with, please solve all acronyms in captions and legends of all Tables and Images (e.g. RBD and NP in figure 1).
Second, Figure 3 and 4 could be improved by a more direct confrontation (e.g. a possible Figure 3a and 3b with current Figure 3 on left side and Figure 4 on the right side). Moreover, in order to be consistent with Table 2, please explain in captions and description of Figures which model is going to be discussed and reported.
According to you estimates, some pre-conditions (e.g. age > 70 yrs, cardiovascular disorders, etc) were identified as protective factors against breakthrough infections in either model 1 or 2. A potential explanation that could be discussed may be that these patients are characterized by higher consistence with preventive measures, and also with higher adherence to vaccine recommendations (with higher vaccination rates) BECAUSE of their higher vulnerability to consequence of COVID-19? Please discuss this topic.
Another topic to be discussed is whether these results may be considered consistent or not with ongoing recommendations about the use of updated monovalent formulations of mRNA vaccines.
Finally, Authors should highlight not only the strengths of their study, but they should similarly stress the limitations (that are reported in final section of the discussion), including the reduced sample size, the substantial self-selection of the paper and mostly the uneven representation of various SARS-CoV-2 strains, that could have affected the eventual estimates.
Comments on the Quality of English LanguageSome minor typos in the results section (e.g. "Receipt of two doses of mRNA-1273 vaccine in the initial; series was not associated with protection from breakthrough infection during 96 weeks; HR 0.862 [0.624, 1.192] or during any COVID-19 wave"). Please double check.
Reviewer 2 Report
Comments and Suggestions for Authors
This manuscript delineates predictors of breakthrough infection in the Canadian population. The methodology and outcomes were sound, reliable and reproducible. However, the revision of the writing quality is required, especially typos. Moreover, the methodology should be clarified and could have all the essential information without reading a further paper.
Major concerns.
1. Suggest adding the inclusion and exclusion criteria to clarify it. The essential brief information must be in the manuscript.
2. Suggest clarifying the criteria for breakthrough infection, including the interval between the last dose and infection. Were some participants infected within less than 14 days after the recent vaccination?
3. Did some participants receive whole-virion inactivated COVID-19 vaccines or COVID-19 vaccines that contained nucleocapsid antigen or encoded gene(s)?
These recipients may alter the anti-nucleocapsid antibody outcomes.
Comments.
1. Line 15 "Vaccines and boosters". It seems unclear what "boosters" meant in this statement.
2. Line 34 " Wuhan ancestral SARS- CoV-2 virus". It is redundant, I suggest using "SARS-CoV-2 ancestral (Wuhan) strain" to clarify it.
3. Line 40 suggests adding "strain" after "ancestral".
4. Lines 43-44 " World Health Organization (WHO) establishing an international antibody standard (binding antibody units-BAU)."
The standardised units for antibodies against SARS-CoV-2 are BAU/mL (binding assay) and IU/mL (blocking assay). Suggests revising it.
5. Line 53: "ancestral and variant SARS- Cov-2 strains ". It is redundant; I suggest using ancestral strain and SARS-CoV-2 variants.
6. Line 65 suggests adding "identifier" before NCT05208983.
7. Suggest re-statement of;
line 131 from "Pfizer-BioTech Comirnaty Omicron BA.4/BA.5" to "Comirnaty Original/Omicron BA.4-5 (Pfizer—BioNTech)"
lines 131-132 from "Moderna Spikevax bivalent BA.4/BA.5" to "Spikevax bivalent Original/Omicron BA.4-5 (Moderna—NIAID)"
8. Suggest clarifying the BMI criteria of "Obesity."
9. Table 3, Booster doses. I wonder if some participants received three (five doses) or four booster doses (six doses).
That is true?
10. Figures 3 and 4. I suggest considering another colour set in the plot to make it colourblindness-friendly.
http://www.cookbook-r.com/Graphs/Colors_(ggplot2)/#a-colorblind-friendly-palette
Typos.
1. Lines 23 and 26. I suggest using "BA.1/2" and "BA.4/5" to make it consistent with the name from the PANGO lineage.
2. Line 50 suggests using "COVID-19" instead of COVID.
3. Line 101 "(COVID)".
4. Lines 99, 104, 230 "COVID".
Comments on the Quality of English LanguagePlease check typos and also "space."
For example, in line 53 "SARS- Cov-2" and "Population- based". There was space between "SARS-" and "Cov-2". Moreover, Cov-2 is also a typo.
Reviewer 3 Report
Comments and Suggestions for Authors
The article presents an interesting analysis of COVID-19 infection in two populations of individuals with two doses of SARS-CoV-2 vaccination. Even though the results have potential clinical use, several points need to be addressed. The abstract does represent the real number of individuals that were analyzed (983 instead of 1286). The second issue is the statistical comparison between two different age groups. It is unclear why the analysis was done by the comorbidities taken by both groups when they should be separate. In addition, the study should have excluded patients with immunomodulatory treatment (transplant patients).
Since the study aims to ascertain the role of antibodies against RBD as protective antibodies, the age groups should be separated. The risk of younger patients being infected is higher and therefore it is expected to be different from the >70 y group. On the other hand, in the >70 y group, comorbidities are a risk of low immune response, and by summing both age groups there is a dilution of the data. The effect is even more complex when patients with cardiovascular disease and diabetes are analyzed.
It is unnecessary to consider Caucasians as a relative risk factor since around 90 % of the participants had that race. The same issue with smokers is a clear type II statistical error. In table 5, what does Age group mean? The table should represent the number of individuals with reinfection and a single infection in the two age groups.
Table 5 also shows the importance of the risk of individuals who were vaccinated with other vaccines that were not mRNA vaccines. It will be crucial to know how many of the Ohter Other groups were vaccinated or not with the bivalent vaccine and how the follow-up of this group was based on RBD antibodies. This is a piece of crucial information as well as the individuals who suffered the viral infection, which had comorbidities. In essence, it is expected that individuals with comorbidities will have a higher risk of natural infection despite having protective antibodies. Another question which was not answered is whether there was more protection depending on the number of booster doses and if there were adverse events.
Recent studies have made it clear that mRNA vaccines boost the cellular memory of individuals with previous SARS-CoV-2 infection. The authors should discuss this issue since memory CD8 response could be important depending on the age group.
Finally, there are two minor issues: 1) there are limitations of the study that should be stated, and 2) can the authors compare their results with the results obtained in other studies? The discussion requires more critical analysis. The authors need to consider the relationship between risk of infection and vaccine protection and obviously neutralizing antibodies represent only one part of the analysis.
Round 2
Reviewer 1 Report
Comments and Suggestions for Authors
Sir,
Authors did a fine job in improving the present paper in accord to my previous comments.
I've no further requests and I'm endorsing its acceptance.
Reviewer 2 Report
Comments and Suggestions for Authors
That is fine for the revision. Good luck with your work.
Reviewer 3 Report
Comments and Suggestions for Authors
The authors have addressed all the points. The manuscript can be accepted.